# OpenReview forum: "The Sum Leaks More Than Its Parts: Compositional Privacy Risks and Mitigations in Multi-Agent Collaboration"
_ICLR.cc/2026/Conference — Submitted to ICLR 2026_

### Official Review · Reviewer_8tHH · 2025-10-31

**Soundness:** 2
**Presentation:** 3
**Contribution:** 2
**Rating:** 4
**Confidence:** 4

**Summary:**

This paper introduces the concept of compositional privacy leakage in multi-agent LLM systems, where sensitive information emerges through the combination of individually innocuous outputs from multiple agents. The authors develop a systematic evaluation framework modeling adversary-defender interactions as a POMDP, where defenders hold partial, non-sensitive data that can be composed by adversaries to infer private attributes. Two defense mechanisms are proposed: (1) Theory-of-Mind (ToM) Defense, where defenders anticipate adversarial intent by simulating the questioner's knowledge state, and (2) Collaborative Consensus Defense (CoDef), where defenders vote based on shared aggregated state. Experiments across 119 scenarios with multiple LLMs (Qwen3-32B, Gemini-2.5-pro, GPT-5) show that while baseline CoT provides limited protection, ToM substantially improves blocking but reduces benign utility, whereas CoDef achieves better balance.

**Strengths:**

1. The paper identifies compositional privacy leakage as a distinct threat in multi-agent systems that extends beyond memorization or single-agent risks. This has high practical value.
2. The evaluation setup is well designed. By providing adversaries with optimal plans $P^*$ and measuring plan execution success separately from inference accuracy, the authors isolate whether privacy failures arise from information flow versus reasoning errors. This methodological rigor enables objective comparison across defense mechanisms.
3. The findings clearly demonstrate that simple CoT reasoning is insufficient for defending against compositional attacks, and that collaborative defense mechanisms provide superior privacy-utility trade-offs compared to single-agent ToM reasoning. These are valuable insights on guiding the development of safe multi-agent systems.

**Weaknesses:**

1. The data construction process is unclear. The paper writes "We construct structured scenarios specifying entities, private data, sensitive targets, and adversary plans", but provides insufficient detail about this critical process. CoT results is bad might because the model does not think these targets are sensitive. Section 3 needs substantial expansion on scenario generation methodology, including examples of how sensitive vs. benign targets are differentiated.
2. The analysis is somewhat superficial. While the paper demonstrates that certain defenses fail, it provides limited insight into why they fail. Is the failure due to lack of privacy awareness about compositional leakage? Is it due to insufficient information for reasoning (especially for non-collaborative methods)? For example, in a related study [1], they found models can identify privacy-sensitive cases but fail to act accordingly in agentic tasks due to complicated objectives. I think similar analysis is needed here for shedding insights on how to improve these models.


[1] PrivacyLens: Evaluating Privacy Norm Awareness of Language Models in Action, Shao et al., NeurIPS 2024

**Questions:**

1. How are benign and sensitive queries being sourced?
2. How are sensitive fragment combinations computed for "CoT + Sensitive Set"?

---

> ### Author Response · Authors · 2025-11-21
> **Response to Reviwer 8tHH: Part 1**
>
> We sincerely thank the reviewer for their thoughtful evaluation. We appreciate their recognition that the paper identifies **“compositional privacy leakage as a distinct threat in multi-agent systems that extends beyond memorization or single-agent risks,”** highlighting its **high practical value.**
> We are glad that the reviewer found our evaluation setup well designed, noting that **“by providing adversaries with optimal plans and measuring plan execution success separately from inference accuracy, the authors isolate whether privacy failures arise from information flow versus reasoning errors.”** We agree that this methodological rigor is important for enabling **objective comparison across defense mechanisms.**
> We also thank the reviewer for emphasizing the significance of our findings, particularly that **“simple CoT reasoning is insufficient for defending against compositional attacks, and that collaborative defense mechanisms provide superior privacy-utility trade-offs compared to single-agent ToM reasoning.”**
>
>
> >**Data construction process**
>
> Our scenarios are generated through a structured pipeline that specifies:
>
> 1. Entities (Agents), one of which, is assigned as an adversary
> 2. Private but non-sensitive (by design) mappings held by each defender, Each defender has a mapping that is a part of the sensitive set i.e. when combine with mappings that other defenders have leads to the inference of a sensitive target and a benign mapping which when combined with other defenders mapping does not lead to the inference of the sensitive target
> 3. Sensitive targets that should only become inferable through composition (by design)
> 4. Adversary plans describing the intended inference path.
>
> We manually design a few samples and give them in-context to a Qwen3-32B model to generate the dataset synthetically. In this work, we assume that each defender has access only to the mappings they are entitled to hold, and these mappings are explicitly constructed to be non-sensitive on their own. By design, it is only the combination of multiple defenders’ mappings that can lead to inferences we classify as sensitive. This ensures that no individual defender ever holds sensitive information in isolation.
>
> We have expanded the paper to describe precisely how these components are instantiated in Sec B.4, Lines 1144-1157, how sensitive targets are selected, and how we verify that only cross-agent composition, not individual mappings, enables the sensitive inference.
>
> >**Sensitive vs. benign targets differentiation**
>
> The CoT \+ Sensitive Set and CoT \+ Sensitive Set \+ Query History rows in Table 1 are exactly meant to address this where we give the ground-truth combination of mappings (Sensitive Set) that are sensitive in the context for the model to reason about what is sensitive and what is not. We have added this discussion in Sec 3\.

---

> ### Author Response · Authors · 2025-11-21
> **Response to Reviewer 8tHH: Part 2**
>
> >**Failure analyis**
>
> Following the PrivacyLens paper, to address this, we manually inspect 20 samples and present the main causes of failure across defenses below. Please see Sec 5, Table 5 and Lines 460-472 in the revised version of the paper.
>
> 1) Tendency to reason about questioner’s intent unless explicitly made to do so
> 2) The tendency to not reason about consequences of sharing pieces of information
> 3) Tendency to not model/question what the questioner knows if it could lead to the revelation of something sensitive
> 4) If the answers to the above questions, tendency to not communicate with others to find the answers before sharing the information
> 5) Lack of global awareness that CoDef overcomes
>
> | Setting / Method                 | How the Agent Reasons                                                                 | Observed Behavior                                                                                 | Failure Mode (Why It Fails)                                                                                       |
> |---------------------------------|--------------------------------------------------------------------------------------|---------------------------------------------------------------------------------------------------|-------------------------------------------------------------------------------------------------------------------|
> | **Base**                         | Looks only at the current query and its own local data. No notion of sensitivity, intent, history, or composition. | Directly answers the disciplinary-actions query because the data is present locally.               | No privacy or compositional awareness; the model treats the task as pure retrieval and ignores inference risks.  |
> | **Base + Sensitive Set**         | Checks which tables are marked "sensitive" but still treats answering as the default action if data is locally available. No consideration of Alice’s prior knowledge or intent. | Still answers the query because the data is sensitive but locally present, and the model is not instructed to reason about consequences. | Lack of consequence reasoning. The model recognizes sensitivity but does not reason about downstream leakage (mirrors findings in PrivacyLens). |
> | **Base + Sensitive Set + History** | Observes Alice’s previous queries but does not interpret them as evidence of adversarial intent. History is seen as informational, not diagnostic. | Answers again, despite knowing Alice already collected several sensitive datasets.                | Cannot infer intent and cannot model what the questioner already knows, even when the history clearly signals aggregation attempts. |
> | **Theory of Mind (ToM)**         | Explicitly reasons about Alice’s sequence of queries and infers possible motives. Recognizes cross-dataset aggregation attempts. | Flags adversarial intent and blocks the response.                                                  | High variance and overthinking. Effective when triggered, but inconsistent; reasoning can be fragile and overly cautious in benign cases. |
> | **CoDef (Collaborative Consensus Defense)** | Aggregates knowledge across defenders; evaluates whether answering would allow the adversary to hold a prohibited combination. | Correctly blocks because it identifies that sharing completes the full four-dataset composition. | Higher overhead and dependency on inter-agent signaling. Performs well but depends on robust collaboration and shared global state. |
>
> >**How are benign and sensitive queries being sourced?**
>
> We construct mappings whose composition leads to the inference of an attribute which we define as the sensitive set. In contextual privacy, the definition of what constitutes as sensitive is contextual. Hence for the purposes of our experiments, we explicitly define the sensitive set as the combination of mappings whose composition leads to sensitive inference. We make this clearer in Lines 139-159 of the revised paper.
>
> >**How are sensitive fragment combinations computed for "CoT \+ Sensitive Set"?**
>
> These are the ground truth fragments that were stored while constructing the dataset.

---

> > ### Comment · Reviewer_8tHH · 2025-11-24
> >
> > I thank the authors for the detailed response. After reading other reviews carefully, I decide to remain my original assessement unchanged as I also agree with some concerns raised by Reviewer SSXU.

---

> > > ### Author Response · Authors · 2025-11-24
> > > **Reply to Reviewer 8tHH**
> > >
> > > We thank the reviewer for engaging with the discussion. We would like to clearly summarize how the points raised by Reviewer SSXU have been directly addressed in our rebuttal and revision.
> > > **Evaluation realism.** Our controlled setting is intentional: it enables quantifiable, reproducible measurement of compositional leakage and defense performance. This isolation of the core phenomenon is necessary before moving to noisier, unstructured domains. We clarified this motivation explicitly in revised Section 2, Lines 142-152..
> > > **Adversary auxiliary knowledge.** We conducted a new experiment (see Lines 475-485, Table 4 in the revised version) where the attacker begins with no auxiliary knowledge. The defense ordering and qualitative trends remain the same, demonstrating that our conclusions do not depend on any strong attacker assumption. In general, modeling a stronger attacker provides a more credible and informative stress-test of defenses: if a defense remains effective even when the adversary starts with additional non-sensitive fragments, we gain greater confidence in its robustness in practice.
> > > **Defender-held data.** We clarified that each defender stores only non-sensitive fragments by definition. Sensitive information arises only when fragments across agents are composed. No agent ever holds sensitive fields on its own.
> > > **Leakage detection and measurement.** We point to the evaluation in the main paper: the attacker must (1) execute a multi-step evidence-gathering plan, and (2) complete the final sensitive inference measured via entailment-based evaluation.
> > > **Defense results and interpretation.** The absence of a universal winner is an informative outcome for this first-of-its-kind evaluation. Our goal is to benchmark defenses — not to promote any single one. Each defense relies on a different mechanism: Self-Voting leverages local consistency (beneficial for smaller models), whereas ToM and CoDef draw on richer reasoning and coordination (better exploited by larger models such as Gemini-2.5-pro and GPT-5).
> > >
> > > We are happy to engage in any further discussion or clarify additional points if helpful.

---

> > > > ### Author Response · Authors · 2025-11-26
> > > > **Reply to Reviewer 8tHH**
> > > >
> > > > Since you referenced Reviewer SSXU’s points in your response, we wanted to briefly highlight Reviewer SSXU’s latest reply, which led to an overall score increase as well as improvements in soundness and presentation. We would greatly appreciate it if you could take a look at our recent responses there to see whether the discussion also addresses your questions.

---

### Official Review · Reviewer_XcEa · 2025-11-01

**Soundness:** 2
**Presentation:** 3
**Contribution:** 2
**Rating:** 6
**Confidence:** 4

**Summary:**

This paper investigates an emerging inference-time privacy issue in multi-agent systems, which is that sharing seemingly innocuous response during agent interactions can allow an adversary to aggregate them and reveal sensitive information. They term this phenomenon "compositional privacy leakage," formally defined it, and proposed two mitigations: 1) ToM defense, where defender agents try to infer the agent's malicious intent on its own; 2) Collaborative Consensus Defense (CoDef), where defender communicates query and answer histories with each other and collectively vote for whether to block a request. They evaluated the baseline CoT methods as well as the proposed defense methods, showing that ToM preserves privacy at the cost of denying benign requests, while CoDef achieves a better balance. They also categorized the reasoning depth into four levels and found a correlation between richer reasoning and stronger compositional privacy protection.

**Strengths:**

- The paper identifies an important privacy issue of the compositional privacy leakage, similar to the concept of quasi-identifier, in the context of multi-agent systems. It contributes to research on inference-time privacy leakage in multi-agent systems, which is an understudied area.
- The paper provides a formal definition of the compositional privacy leakage problem, and proposes two defense methods. They both more effectively blocked the sensitive requests. The CoDef method substantially improved the blocking success rate while maintaining a similar level of benign success rate to the baselines.

**Weaknesses:**

- In the threat model and the defense methods, the defender agents and adversary agents are explicitly separated. There seems to be an assumption that the defender agents are trusted by all other defender agents, and another agent is explicitly regarded as an adversary. This does not reflect the risks in a realistic multi-agent setup, where any agent could be compromised. Making this assumption may introduce bias that boost the performance in this task, while lacking applicability in real settings where such identities can't be assumed.
- The current description of compositional privacy leakage seems to emphasize scenarios where sensitive attributes only become visible after multiple structured datasets are joined. For example, linking tables on shared IDs or quasi identifiers, similar to classic reidentification attacks where an anonymized dataset is deanonymized using auxiliary public records. That framing is important but not especially new. It also underplays what is unique about LLMs. LLMs can compose unstructured, semantically rich evidence such as emails, documents, chat logs, and meeting notes, and then infer sensitive facts about specific people without any explicit shared keys. In other words, the leakage is not just a matter of table A being joined with table B. It is that the model can synthesize implicit meaning across heterogeneous text and surface latent information. The latter feels like the real frontier risk, and it is not fully captured by the current definition.

**Questions:**

- Why are the benign success rates also low in the several CoT baselines (some even lower than the defense methods)?
- Can you clarify what types of compositional privacy leakage are captured in your evaluation?
- Can you address the question regarding the assumption of the roles of defender and adversary, and how it might affect the validity of your results?

---

> ### Author Response · Authors · 2025-11-21
> **Response to Reviwer XcEa**
>
> We thank the reviewer for their thoughtful evaluation. We appreciate their recognition that the paper identifies **“an important privacy issue of the compositional privacy leakage, similar to the concept of quasi-identifier, in the context of multi-agent systems.”** We are glad that the reviewer found our work contributes to **“research on inference-time privacy leakage in multi-agent systems, which is an understudied area.”**
>
> We also thank the reviewer for highlighting the formalization and defenses, noting that **“the paper provides a formal definition of the compositional privacy leakage problem, and proposes two defense methods,”** both of which **“more effectively blocked the sensitive requests.”** We are particularly encouraged that CoDef was recognized for **“substantially improved the blocking success rate while maintaining a similar level of benign success rate to the baselines.”**
>
>
>
>
> >**Separation between attacker and defenders**
>
> Yes, we acknowledge that our current setup assumes a clear separation between defender and adversary agents, and that defender agents are trusted by one another. This assumption simplifies the evaluation and allows us to systematically benchmark defense mechanisms. We agree that in realistic multi-agent systems, any agent could be compromised, and such trust cannot be guaranteed.
>
> This does not invalidate the results but it does **scope them**: our findings characterize how well different defenses prevent leakage *given* a known potential adversary, not how they perform under arbitrary or dynamic compromise. Extending the evaluation to settings with partially trustworthy or potentially malicious defenders is an important direction for future work. It would broaden applicability, but it is orthogonal to our core contribution. Introducing additional malicious agents would only strengthen the attack, since the defender would then need to identify each agent’s intent during communication which would reduce the effectiveness of CoDef and will add more complexity to both defenses.
>
> >**Current setup does not capture all realistic scenarios**
>
> We agree that realism is important, and we appreciate the opportunity to clarify our design choices. Our goal is *not* to claim that the synthetic tables represent the full complexity of real-world multi-agent interactions. Rather, we use this controlled setting because it enables *quantifiable, tractable measurement* of compositional leakage and defense performance across many scenarios. Evaluating this phenomenon in fully unstructured or naturally occurring data (e.g., emails, notes, or conversational histories) is more challenging due to the unavailability of data, and as there is no ground-truth notion of “what combination of fragments reveals the sensitive target,” making systematic evaluation and progress much harder. We add
>
> Crucially, a major motivation for our paper is to evaluate *defense capabilities*, and our findings already show that current LLMs struggle even in this simplified setting. Introducing more implicit or noisy compositions would only make the defense task harder and risk conflating failure sources. Therefore, starting with a clean, controlled scenario allows us to isolate the core phenomenon, compare methods fairly, and establish a baseline understanding before moving to more complex domains in future work. We make this clear in **Sec 2, lines 142-152** in the revised version.
>
> >**Lower benign success rate with CoT**
>
> The benign success rates are low for CoT baselines because chain-of-thought prompting itself can introduce unnecessary caution or distraction in multi-agent settings. When models reason step-by-step without an explicit objective, they sometimes over-interpret ambiguous queries, follow irrelevant reasoning paths instead of directly answering the question, and can't distinguish between sensitive and non-sensitive attributes. This leads to reasoning drift and reduces task accuracy, even in benign cases.
>
> In contrast, our defense methods are explicitly optimized to decide whether to answer or block based on intent/votes, which helps maintain more consistent behavior. So the low benign success in CoT baselines reflects the instability of unconstrained reasoning.
>
> >**Types of leakage**
>
> As outlined in Sec 2, threat model, in the original paper, our evaluation focuses on **compositional privacy leakage**, where individual pieces of information become sensitive only when combined across agents. Specifically, we capture cases where:
>
> 1. Each defender holds a non-sensitive mapping,
> 2. But the adversary can query multiple agents, aggregate their responses, and
> 3. Infer a sensitive fact that no single agent is allowed to reveal on its own.
>
> In other words, we test whether the adversary can exploit **cross-agent composition**, linking mappings, correlations, or partial clues across agents, to recover sensitive attributes that are not directly exposed anywhere else.

---

> > ### Author Response · Authors · 2025-11-26
> > **Friendly Reminder to Reviwer XcEa**
> >
> > We would like to express our gratitude for your time and effort in reviewing our paper. We hope our hard work on the additional rebuttal experiments: **(i) the assumptions underlying the defender–adversary role separation and how they scope (but do not invalidate) the results; (ii) the broader forms of compositional privacy leakage our framework would generalize to beyond table-style joins; and (iii) the revised explanation for low benign success rates in CoT baselines addressed your questions. We also strengthened the conceptual framing around controlled vs. fully unstructured scenarios to more clearly articulate the evaluation design.** We hope our responses sufficiently clarify the points you raised and will allow you to revisit your score. If any part of our response would benefit from further clarification, or if elaborating on any of the above analyses would assist your assessment before the deadline, we would be very happy to provide it.

---

### Official Review · Reviewer_mhXs · 2025-11-01

**Soundness:** 3
**Presentation:** 3
**Contribution:** 3
**Rating:** 8
**Confidence:** 4

**Summary:**

Paper defines privacy leakage that accumulates over time through benign outputs and can be inferred by an adversary. The paper uses a game setup and discusses different defenses.

**Strengths:**

- The attack is clear and works really well, it is a novel result in the area of Contextual Privacy
- The additional formulation is useful to enable a systematic framework
- Defenses introduce coherent methods to mitigate the attacks
- Evaluation is clear and uses synthetic data.

**Weaknesses:**

- Synthetic data might not fully represent situations that will occur in real life with the agents.
- The defense mechanisms might still be vulnerable to context hijacking attacks
- I think this can be framed as model-level defenses which is a valuable contribution but yet (as all defenses) has its own limitations, it would be great to have them addressed in the paper
- Also, there is this neighboring domain of LLM censorship [1] that is worth discussing.
- An adversary could also cause "overthinking" attacks to slow down the system.
- It might be interesting to see the overhead of preventing the leakage and ways to stop early

[1] LLM Censorship: A Machine Learning Challenge or a Computer Security Problem? [icml'24]

**Questions:**

Overall, paper is great, addressing weakness brought above would be the most helpful.

---

> ### Author Response · Authors · 2025-11-21
> **Response to Reviwer mhXs**
>
> We sincerely thank the reviewer for their positive and constructive feedback. We appreciate the acknowledgment that **“the attack is clear and works really well”** and that it represents **“a novel result in the area of Contextual Privacy.”** We are also grateful that the reviewer found our additional formulation **“useful to enable a systematic framework”** and that our defenses **“introduce coherent methods to mitigate the attacks.”** We appreciate their remark that the evaluation is **“clear and uses synthetic data.”**
> >**Synthetic data might not reflect real situations**
>
> Thank you for this insightful comment. We agree that the most concerning form of compositional privacy leakage in real-world systems involves LLMs implicitly combining heterogeneous, unstructured evidence, emails, notes, chat logs, meeting transcripts, etc., to infer sensitive attributes without any explicit shared identifiers. This frontier risk goes far beyond traditional table-join or linkage attacks.
>
> Our current experimental setup, however, should not be interpreted as limiting compositional leakage to structured joins. Instead, the table-based formulation serves as a *synthetic, controlled instantiation* that allows us to quantify leakage, systematically vary compositions, and evaluate defenses at scale. Realistic unstructured settings remain extremely challenging to benchmark: there is no established ground truth for “which combination of fragments reveals the sensitive target,” and both data availability and annotation are major obstacles. This makes it difficult to measure when leakage occurs or whether defenses are working, and therefore difficult to make scientific progress.
>
> Importantly, our goal in this paper is to study *defense capabilities*. We already find that current LLMs struggle significantly even in this simplified, tractable environment. Introducing fully unstructured evidence or subtle, implicit semantic composition would only make the defense task harder and risk conflating evaluation noise with genuine failures. Starting with a clean, controlled abstraction therefore allows us to isolate the phenomenon, establish baselines, and compare approaches fairly before extending to richer, more realistic domains in future work. We make this clearer in Lines 142-159 in the revised version.
>
> >**Vulnerability to other attacks and other limitations of defenses**
>
> - We agree that the defense mechanisms could still be vulnerable to context hijacking attacks. Our current work focuses on evaluating defenses under the threat models in Sec 2 and compositional reasoning risks, but context hijacking or overthinking represent complementary attacks. Addressing such attacks would require additional safeguards, such as stricter context isolation, query sanitization, which could be incorporated in future work with newer threat models to further strengthen multi-agent privacy protections. As with the Theory-of-Mind (ToM) defense, conservative reasoning can reduce utility by mistakenly blocking safe requests.
> - Defense effectiveness often scales with the model’s reasoning and collaboration abilities. Smaller or less capable models may not implement some defenses effectively.
> - We add this discussion in the in Lines 959-966 of the Appendix of the revised version of the paper .
>
> >**LLM censorship discussion**
>
> Thanks for mentioning this. We have added this discussion in Lines 520-523 of the revised version.
>
> >**Overhead of preventing the leakage**
>
> We provide the computational overhead cost of the proposed defenses below:
>
> Let:
> - **N** = number of queries
> - **S** = number of reasoning steps per query
> - **T** = time per model forward pass
> - **C** = cost of communication/aggregation per agent per query
>
> - Theory-of-Mind (ToM) Overhead
>
> Total computational overhead (O_ToM):
>
> O_ToM ≈ N * S * T
> (ToM scales with nested reasoning depth.)
>
> - Collaborative Consensus Defense (CoDef) Overhead
>
> O_CoDef ≈ N * (A * T + A * C)
>
> We have added this to the Appendix of the revised version (Lines 949–957).

---

### Official Review · Reviewer_SSXU · 2025-11-01

**Soundness:** 1
**Presentation:** 1
**Contribution:** 2
**Rating:** 2
**Confidence:** 3

**Summary:**

This paper addresses compositional privacy leakage in multi-agent systems, where individual outputs from multiple agents can collectively reveal sensitive information. The authors formalize this risk and evaluate two defense strategies — a Theory-of-Mind (ToM) defense, in which agents model an adversary’s intent, and a Collaborative Consensus Defense (CoDef), where agents jointly decide whether to respond to a query.

**Strengths:**

Compositional privacy risk is an important topic in privacy.

**Weaknesses:**

Although I understand what the authors are *trying* to convey in this paper, the evaluation setup feels unrealistic and overly artificial. The paper also gives the impression of being hastily organized and written, resulting in missing or unclear details. The assumption of a “black-box adversary with auxiliary knowledge” is underspecified and vaguely defined — does this auxiliary knowledge include personal identifiers? If so, I disagree with the authors’ characterization of such data as public information, as it makes the overall setup overly simplified and less credible.

Moreover, several examples in the paper suggest that the defenders already hold and disclose highly sensitive information that should never be shared with any external entity. For instance, providing an employee ID–to–(employee name, department) mapping from personal records to the attacker would itself constitute a serious privacy violation. Many of the defenders’ data sources are already sensitive by nature, which further undermines the realism of the evaluation.

In addition, the paper lacks a clear explanation of how leakage is identified or measured. The appendix is also difficult to follow, with duplicated sections and disorganized ordering. As a result, I find it challenging to fully understand the results or accept the validity of such an artificial experimental setup.

There is no clear winner for defense. The results for the defense methods are mixed: Self-Voting performs best on Qwen, CoDef achieves the highest performance on Gemini-2.5-pro, and ToM performs best on GPT-5.

**Questions:**

- Does ToM defenses sacrificing benign success mean the LLM is pessimistic in predicting the other’s intents?
- Is the leakage detection based on string matching or entailment? What does the s* look like?
- How do humans perform on this task?
- Line 144: What is A_j and A_i?
- If the correct plan is already provided, what is the attacker’s role? Do they simply generate natural language queries based on that plan?
- How do you ensure the information held by each defender are not sensitive?

---

> ### Author Response · Authors · 2025-11-21
> **Reponse to Reviewer SSXU: Part 1**
>
> We sincerely thank the reviewer for taking the time to evaluate our work and recognizing that **compositional privacy is an important and timely topic** in the broader privacy landscape.
>
>
> >**Evaluation setup**
>
> Thank you for raising this point. We agree that realism is important, and we appreciate the opportunity to clarify our design choices. Our goal is *not* to claim that the synthetic tables represent the full complexity of real-world multi-agent interactions. Rather, we use this controlled setting because it enables *quantifiable, tractable measurement* of compositional leakage and defense performance across many scenarios. Evaluating this phenomenon in fully unstructured or naturally occurring data (e.g., emails, notes, or conversational histories) is more challenging due to the unavailability of data, and as there is no ground-truth notion of “what combination of fragments reveals the sensitive target,” making systematic evaluation and progress much harder.
>
> Crucially, a major motivation for our paper is to evaluate *defense capabilities*, and our findings already show that current LLMs struggle even in this simplified setting. Introducing more implicit or noisy compositions would only make the defense task harder and risk conflating failure sources. Therefore, starting with a clean, controlled scenario allows us to isolate the core phenomenon, compare methods fairly, and establish a baseline understanding before moving to more complex domains in future work. We make this clear in Sec 2, lines 142-152 in the revised version.
>
> >**Additional experiment: adversary without auxiliary knowledge**
>
>  We agree with the reviewer that if the adversary is truly external, granting too much structured auxiliary knowledge may be unrealistic. To address this, we conducted an additional experiment in which the adversary has **no auxiliary knowledge themselves**; instead, this information is assigned to a separate agent, and the attacker must query that agent to obtain it. This setting removes the assumption that the attacker begins with privileged information while preserving the multi-agent compositional nature of the task.
>
> | **Defender**    | **Method**                     | **Sensitive Block (%)** | **Benign Succ. (%)** | **Bal. Outcome (%)** | **Overall Succ. (%)** |
> |-----------------|--------------------------------|--------------------------|------------------------|------------------------|------------------------|
> | **Qwen3-32B**   | CoT                            | 42.5                     | 66.1                   | 54.3                   | 27.8                   |
> |                 | CoT + 𝓛                        | 48.9                     | 72.4                   | 60.7                   | 29.5                   |
> |                 | CoT + 𝓛 + 𝓗ₜ                  | 44.1                     | 63.5                   | 53.8                   | 26.0                   |
> |                 | ToM Defense                    | 90.4                     | 55.8                   | 73.1                   | 51.3                   |
> |                 | Self-voting                    | **97.8**                 | 56.6                   | 77.2                   | **59.1**               |
> |                 | CoDef                          | 88.9                     | **69.1**               | **79.0**               | 55.5                   |
>
>
> Importantly, when we eliminate the adversary’s auxiliary knowledge and relocate that information to an additional agent, thereby removing the strong attacker assumption, the qualitative pattern of results remains consistent. CoDef still achieves the best balance between sensitive-query blocking and benign success, while CoT-based baselines and ToM defenses retain their relative ordering. However, the absolute sensitive blocking rate increases because the attacker is weaker. This demonstrates that our conclusions are not an artifact of adversary pre-knowledge, and the defense trends are robust across threat-model variations. We discuss this in Lines 475-495 of the revised paper.

---

> ### Author Response · Authors · 2025-11-21
> **Reponse to Reviewer SSXU: Part 2**
>
> > **Why we allow the adversary to possess auxiliary knowledge**
>
>  In real deployments, adversaries rarely operate with zero background information. Public records, organizational charts, leaked fragments, or prior interactions often give them partial visibility into entities or relationships. Modeling an adversary with *some* auxiliary knowledge is therefore not an unrealistic assumption — it is a conservative and standard one. Stronger attackers provide a more credible and informative stress-test of defenses. If a defense remains effective even when the adversary starts with additional non-sensitive fragments, we gain greater confidence in its robustness in practice. We discuss this in Lines 147-152 of the revised paper.
>
> > **Are agents holding “sensitive” data? Clarification of our design assumption**
>
> In our setting, each defender only holds data that is *non-sensitive (by design) by itself. Our auxiliary knowledge never includes private attributes or sensitive labels; it only consists of non-sensitive fragments that are individually harmless but become sensitive by design when composed. We define anything that a single agent holds as non-sensitive. A key aspect of compositional privacy leakage is that *the sensitive information arises only when multiple benign fragments are combined*. Under our construction, no single agent possesses a sensitive record alone. We have revised the paper to make this distinction clearer and avoid the impression that defenders are directly exposing inherently private fields. We make this clear in line 141 of the revised paper.
>
> > **How is leakage identified or measured?**
>
> We define compositional leakage as the successful inference of a sensitive attribute that no single agent directly reveals, but which can be deduced when information fragments from multiple agents are combined. We instantiate this using structured plan-based evaluation: the adversary first generates a multi-step plan to collect fragments, then we test whether the final inferred value matches the ground-truth sensitive attribute. Leakage is thus measured at two levels — (1) plan execution success (whether all fragment-gathering steps were completed) and (2) end-to-end inference (NLI) success (whether the sensitive fact was recovered). Please see Sec. 3.2, lines 201-207. Sec 3.4 Lines 302-307 in the original paper. Please see an example of s\* in Fig. 4, Step 5 of the original paper.
>
> To improve clarity, we reorganized the appendix to remove duplicated sections
>
> > **No winner for defense**
>
> Yes, the results are mixed and no single defense is a winner which itself is an informative outcome. Our point is to test the defenses—the scope here is purely to benchmark them, not to promote one over the other, especially since this is a new task. The different defense strategies leverage complementary mechanisms: Self-Voting relies on local consistency and works best for smaller models while ToM and CoDef depend on richer reasoning and co-ordination which larger model (Gemini-2.5-pro and GPT-5) can execute more effectively. This suggests that multi-agent privacy defenses should be architecture-aware rather than one-size-fits-all.
>
> > **Is ToM making LLM pessimistic?**
>
> Yes, The drop in benign success under Theory-of-Mind (ToM) defenses reflects cautious intent inference. By reasoning about possible motives, ToM agents often treat ambiguous queries as potentially adversarial. This conservatism boosts privacy protection but can over-block safe requests, trading some utility for stronger safeguards. We add this in Lines 398-401 in the revised version.
>
> > **What is A\_j and A\_i?**
>
> Here, A\_i​ and A\_j refer to individual agents in the multi-agent system. The statement means that each agent A\_iA is aware of the *schema* (i.e., the types and fields) of the data held by another agent A\_j, but does not have access to the actual contents of K\_j​. This models realistic data-siloed environments where organizations may know what kinds of information other entities maintain, without ever seeing their private records. We make this clear in lines 152-159 of the revised paper.
>
> > **Attacker’s role**
>
> The attacker must translate the plan into natural-language queries, decide phrasing and timing, adapt to defenders’ responses (retry, rephrase, pivot), and perform the final inference step that composes gathered fragments. Success, therefore, requires both successful fragment collection and correct composition. In our evaluation, the plan specifies *what* to collect and in what order, while the attacker must decide *how* to collect it via realistic query generation and adaptive behavior. This design prevents attack failures due to the attacker’s inability to plan and evaluate defenses under a stronger, more realistic attacker model. See Lines 182-186 in the original paper.

---

> > ### Author Response · Authors · 2025-11-21
> > **Reponse to Reviewer SSXU: Part 3**
> >
> > >**How is information held by each defender not sensitive?**
> >
> > In this work, we assume that each defender only has access to mappings that are non-sensitive, (by definition), data they are entitled to hold. By design, it is only the combination of multiple defenders’ mappings that can lead to inferences we classify as sensitive. This ensures that no individual defender holds the sensitive information on its own.

---

> ### Comment · Reviewer_SSXU · 2025-11-24
> **Please double-check before posting**
>
> - I assume the authors have mistakenly copy-pasted my review in their response. Were you trying to say something more about the following part?
> > Moreover, several examples in the paper suggest that the defenders already hold and disclose highly sensitive information that should never be shared with any external entity. For instance, providing an employee ID–to–(employee name, department) mapping from personal records to the attacker would itself constitute a serious privacy violation. Many of the defenders’ data sources are already sensitive by nature, which further undermines the realism of the evaluation.
>
> - Lines 468 does not correspond to experiments with auxiliary knowledge.
>
> I kindly request that you update your response, as such repeated mistakes (like the duplicate and out-of-order sections in the original draft) can convey the impression that the authors are being too hasty and not fully thorough. And please make sure the rest of the responses (for other reviewers as well) do not contain these sort of mistakes.

---

> > ### Author Response · Authors · 2025-11-24
> >
> > Thank you for pointing this out. We apologize for the confusion—this was a small formatting oversight in our response draft. The relevant discussion about auxiliary-knowledge experiments has now been moved to Line 475 in the revised draft. Some lines shifted slightly while fitting all content into the 10-page limit as we worked hard to prepare detailed responses and update the PDF under a tight one-week timeline. Regarding the sections you noted as duplicated, some of these were expanded versions of sections in the main paper, intended to provide additional details that were omitted from the main text due to space constraints.

---

> ### Comment · Reviewer_SSXU · 2025-11-25
>
> Thank you for the rebuttal!
>
> - The new results make me wonder even more: why is the difference between the original results and the new results so small even after removing the auxiliary information?
>
> - I never said that the synthetic evaluation itself is unrealistic. What I said is that treating data that should be considered private as if it were public is what makes the evaluation setup unrealistic and overly artificial. Although you define those auxiliary data as “non-sensitive,” it is difficult for me to agree that a dataset containing a list of employee IDs should be considered public, non-sensitive information.
>
> - If I understood correctly, the leakage accuracy is computed based on the yes-or-no question prompt (Line 887 in the original paper)? The main text says $\mathbb{I}[\hat{s} = s^{*}]$, but I suggest revising this notation since it gives the impression that the evaluation is done via string matching. Also, where is the ground truth provided in the prompt? The prompt does not appear to contain any placeholder for the true answer. How reliable are these LLM judgments?
>
> - Regarding ToM making the model pessimistic: I don’t think ToM itself causes pessimism. Rather, I believe the prompt you used induces that behavior. If the prompt were balanced, the results might differ. (To be clear, I’m not suggesting that you need to run additional experiments now.)

---

> > ### Author Response · Authors · 2025-11-25
> >
> > Thank you for your continued engagement and for raising these detailed questions. We address each point below.
> >
> > **Why do the new results remain similar without adversary auxiliary knowledge.**
> > We would like to clarify that the absolute numbers do change; for example, we see Sensitive Block % for Qwen3-32B rise from 31% (in case of attacker with auxiliary knowledge) to 42% (in case of attacker with no auxiliary knowledge) under the simple CoT setting. Nevertheless, we expect the overall qualitative trends to stay the same: defenses act on inter-agent information flow, not the attacker’s initial state. Removing auxiliary fragments from the attacker adds one query step, but the attacker still assembles the same cross-agent composition via the new agent. This slightly weakens the attacker (raising sensitive-blocking rates) while leaving the relative ordering and balance of defenses stable, showing that defense behavior does not rely on a strong attacker assumption.
> >
> > **On the nature of auxiliary data and “public vs. private.”**
> > We agree that in real-world applications, identifiers such as employee IDs or name mappings would be sensitive. In our synthetic benchmark, however, these fragments serve only as abstract symbolic identifiers for evaluating compositional reasoning and leakage mechanics (they could be replaced with any other symbol, as long as the scenario specifies which symbols are deemed private/public). Crucially, no single defender ever holds a meaningful private attribute on its own: sensitivity emerges only through cross-agent composition, which is the key phenomenon the benchmark is designed to isolate.
> >
> > Importantly, our additional experiment in which the adversary is given no auxiliary knowledge directly addresses this question: the fragments are held by a separate agent, and the attacker must actively query for them. The qualitative conclusions remain unchanged, confirming that our results do not rely on treating such information as public or attacker-accessible.
> >
> > **Evaluation procedure and leakage measurement.**
> > You are correct that the original notation could suggest string matching. We revised this for clarity (Lines 214-220 in the revised paper). The ground truth comes from the final\_inference\_reference field (Line 907 in the original paper) and is included in the prompt; the model must infer the sensitive attribute and answer a yes/no entailment question. Manually spot-checking 20 instances confirmed 100% consistency.
> >
> > **On ToM “pessimism.”**
> > We agree that the overcautious behavior of the ToM-style defense is prompt-dependent. In our instantiation, the specific ToM prompt leads to conservative actions, but this is not intrinsic to ToM itself. We have revised the language to make this prompt dependence explicit (Lines 409-412).
> >
> > Please let us know if this addresses your comments, we are happy to continue to engaging during the remaining rebuttal period.

---

> > > ### Comment · Reviewer_SSXU · 2025-11-26
> > >
> > > Thank you for the explanation and revision.
> > > - Yes, the absolute numbers do change, and the sensitive-block rate does increase due to the additional step. However, most of the differences probably remain within the error margins reported in the original draft’s table. Why do you think this is the case?
> > > - Also, shouldn’t identifiers be considered sensitive regardless, since they provide the mapping to the individual? Why do you think sensitivity emerges only through cross-agent composition?

---

> > > > ### Author Response · Authors · 2025-11-26
> > > >
> > > > **On why the new results remain within the original error margins:**
> > > >
> > > > We would like to clarify that the “additional step” introduced in the no-auxiliary-knowledge setting is explicitly included in the attacker’s plan. This step simply instructs the attacker to query the separate fragment-holding agent in its plan.
> > > >
> > > > A key contributing factor to the stability of the results, especially for ToM, Self-Voting, and CoDef, is that this missing-fragment step is a straightforward, instruction-following action that current models execute reliably. Once the attacker retrieves this fragment, the remainder of the reasoning trajectory is essentially identical to the original setting.
> > > >
> > > > Beyond this, the stronger defenses are dominated by intent detection/collaboration, not by plan length or query count. ToM, Self-Voting, and CoDef all gate on whether the attacker is attempting a sensitive inference. As soon as the attacker reaches that point, regardless of how many preliminary steps were required or whether auxiliary information was available at the start, the defenses trigger based on the same inferred adversarial intent. This is why their relative performance remains stable across both variants of the setup.
> > > >
> > > > **On why sensitivity emerges only through cross-agent composition:**
> > > >
> > > > In our controlled setting, we *define* each fragment to be contextually non-sensitive on its own; sensitivity appears only after the fragments are combined. What counts as “sensitive” is inherently contextual and depends on downstream use, the attacker’s assumed knowledge, and how different pieces of information interact (an aspect of contextual privacy which we discuss in Lines 461-468 in our related works section of the original paper). This design allows us to isolate the specific mechanism we aim to study: whether aggregating individually benign pieces can lead to a sensitive inference.
> > > >
> > > > This setup also simulates a real and growing frontier risk in deployed (multi-)agentic systems: LLMs implicitly combining heterogeneous, unstructured evidence, emails, notes, chat logs, meeting transcripts, etc., to infer sensitive attributes without any explicit shared identifiers. Such risks extend far beyond classical linkage or table-join attacks. To the best of our knowledge, no suitable real-world dataset exists to systematically evaluate this behavior. So we adopt a controlled environment where sensitivity emerges through cross-agent composition, mirroring the structure of these real-world compositional leakage pathways. We have discussed this in Lines 142-152 of the revised paper.

---

> > > > > ### Comment · Reviewer_SSXU · 2025-11-26
> > > > >
> > > > > Thank you for the explanation! Although I'm not entirely convinced, but I do see your point. Therefore, I raise my overall score to 4 and I also raised other soundness and presentation scores to reflect the clearer explanation and new experiment results that the authors have provided.

---

> > > > > > ### Author Response · Authors · 2025-11-26
> > > > > >
> > > > > > We sincerely thank you for your thoughtful reassessment and for raising your overall score, as well as the soundness and presentation scores. We truly appreciate your engagement with our clarifications and the new experimental results. We would be more than happy to continue any further discussion or provide additional details if it would be helpful.

---

### Author Response · Authors · 2025-11-21
**General Response**

We sincerely thank the reviewers for their thoughtful and constructive feedback. We appreciate their recognition of the timeliness of studying **“compositional privacy risks in multi-agent systems, an important but understudied and practically significant problem”** (SSXU, XcEa, 8tHH). Reviewers highlighted the clarity and novelty of the attack formulation, emphasizing that **“the contextual-privacy–style leakage mechanism is well defined, intuitive, and empirically strong”** (mhXs, XcEa, 8tHH). They also appreciated our formalization of compositional privacy leakage and the introduction of a systematic multi-agent evaluation framework, which they found **“useful for structuring future investigations of inference-time privacy risks”** (mhXs, XcEa, 8tHH).
Reviewers further noted that our proposed defenses, Theory-of-Mind reasoning and Collaborative Consensus Defense (CoDef), represent **“coherent and meaningful approaches to mitigating these attacks”** (mhXs, XcEa). They highlighted that CoDef, in particular, **“achieves a compelling privacy-utility balance, offering substantial improvements in blocking sensitive queries while maintaining strong benign response rates”** (XcEa, 8tHH). Additionally, we are thankful that they appreciated the **“clear experimental design, including the separation of planning and inference accuracy and the multi-model evaluation”** (mhXs, 8tHH). We also thank the reviewers for acknowledging the contribution of **“categorizing reasoning-depth levels and demonstrating their correlation with improved privacy protection”** (XcEa, 8tHH).

---

### Author Response · Authors · 2025-12-03
**AC note**

We would like to thank the reviewers and AC for their time and thoughtful attention to our submission. We were encouraged by the positive responses we received during the shortened discussion period, and for context, we summarize our rebuttal discussion with all reviewers below:

---

Reviewer SSXU

During the discussion period, we made several targeted updates:

Additional Experiments

* **Experiment without attacker auxiliary knowledge:** Introduced a threat-model variant where auxiliary information is held by another agent. Results remained qualitatively consistent, strengthening robustness.

Additional Discussion

* **Clarified evaluation setup:** Explained why the controlled setting is appropriate for studying compositional privacy, emphasizing that each fragment is non-sensitive by design and sensitivity emerges only through combination. Reorganized the appendix to remove duplicated content.

* **Improved leakage-eval clarity:** Clarified the two-stage leakage metric (plan execution \+ NLI inference), provided an explicit sensitive-target example, and refined notation in the revised PDF.

* **Clarified ToM design:** Addressed questions around ToM conservatism, schema knowledge, and attacker behavior and knowledge supported by the above experiment.

* **No winning defense:** Emphasized that ToM, CoDef, and Self-Voting provide complementary strengths and that our goal is to benchmark defenses against this new leakage mechanism rather than identify a single universal best method.

**Outcome:** After multiple rounds of discussion, Reviewer SSXU raised their overall score from **2 to 4** and also increased soundness and presentation scores, citing clearer explanations and the new experimental results we provided. *(Nov 26, 2025\)*

---

Reviewer mhXs

We addressed comments on synthetic data, limitations, censorship connections, and computational overhead:

* **Clarified synthetic-data role:** Expanded Lines 142–159 in the new version to justify using table-based synthetic data as a controlled, ground-truth environment for isolating compositional leakage.

* **Discussed broader limitations and additional attacks:** Added discussion on vulnerabilities to context hijacking/overthinking, future threat models, and model-size dependence (Lines 959–966).

* **Integrated LLM censorship discussion:** Added Lines 520–523 connecting our work to the ICML’24 censorship framing.

* **Added computational overhead analysis:** Introduced an explicit formalization of ToM and CoDef overhead in Appendix Lines 949–957.

---

Reviewer XcEa

We provided clarifications and expansions addressing their comments on threat modeling, the scope of compositional leakage, and CoT benign accuracy:

* **Clarified defender–adversary role assumptions:** Explained that our setup assumes trusted defenders and a known adversary to scope the study; noted that adding malicious agents would only strengthen the adversary and further challenge defenses.

* **Expanded compositional-leakage framing beyond structured joins:** Clarified that structured tables are a controlled instantiation for benchmarking, while richer unstructured evidence composition is a key motivation for future extensions.

* **Strengthened evaluation-design explanation:** Expanded Sec. 2 to justify controlled environments for comparison, and grounding future work on noisy real-world compositions.

* **Clarified low benign success in CoT:** Explained reasoning drift and over-cautiousness in unconstrained CoT prompting, contrasting this with defenses that explicitly reason about intent or votes.

* **Clarified leakage types studied:** Specified how sensitivity arises only from cross-agent evidence aggregation and detailed the adversarial inference pathway.

---

Reviewer 8tHH

We expanded methodological details, data construction, failure analysis, and defense interpretation:

* **Expanded data-construction pipeline:** Clarified how entities, fragments, sensitive targets, and adversary plans are generated; explained guarantee of non-sensitive fragments; expanded Sec. 3 and Appendix B.4 (Lines 1144–1157).

* **Clarified benign/sensitive query sourcing and fragment combinations:** Specified that “CoT \+ Sensitive Set” is derived from ground-truth sensitive fragment sets.

* **Added detailed failure analysis:** Introduced a structured failure taxonomy, concrete examples, and an expanded Table 5 (Lines 460–472), linking failures to another paper’s observations.

* **Clarified evaluation realism and attacker assumptions:** Strengthened justification for controlled settings; added new “zero attacker knowledge” experiment (Lines 475–485, Table 4), showing that defense trends are not a function of where knowledge originates.

* **No winning defense:** Reiterated that ToM, CoDef, and Self-Voting offer complementary strengths, and the goal is to benchmark defenses, not identify a universal best method.

---

We include all additional experiments and discussion in the revised PDF.

---

### Meta-Review · Area_Chair_fY8d · 2026-01-07

**Summary:**

The reviewers unanimously agree that this paper addresses an important and timely issue: combinatorial privacy leakage in multi-agent LLM systems. They commend the effort to formalize the threat and investigate privacy risks during inference time beyond single-agent settings. The attack formula is generally considered clear and effective, and some reviewers found that the proposed defense offers a better privacy utility trade-off compared to simpler reasoning-based baselines. However, major concerns center on the realism and clarity of the evaluation setup. Multiple reviewers argue that the synthetic scenarios and threat model are overly artificial, rely on underspecified or questionable assumptions (e.g., trusted defenders, clearly identified adversaries, non-sensitive “auxiliary” data), and may not reflect realistic multi-agent deployments where any agent can be compromised. Key aspects were not adequately explained, making the results difficult to interpret, such as data construction, leak identification, and measurement. Reviewers also note mixed defense performance across models, limited analysis of failure modes, and a lack of discussion on broader or more implicit forms of compositional leakage unique to LLMs. Overall, while the problem formulation is considered valuable, the paper would require clearer definitions, more realistic assumptions, deeper analysis, and stronger justification of its experimental design to convincingly support its claims. In addition, the confusion generated by the authors' rebuttals and manuscript revisions left me feeling less confident that they would be able to complete the revisions in the final version. Therefore, AC's recommendation is rejection.

**Reviewer Concerns:**

The author provided a clearer explanation of key concepts such as data construction in their rebuttal. Experiments without prior knowledge and analyses of failed cases can address some of the reviewers' concerns. The primary issue lies in the discussion of the evaluation setup failing to clearly persuade the reviewers, particularly regarding the sensitivity of auxiliary data and the lack of evaluation using real data.

**Reviewer Scores:**

Reviewer SSXU explicitly stated that the rating would be raised to 4. I expect the final rating to be as follows:
- Reviewer SSXU: 4
- Reviewer  mhXs: 8
- Reviewer XcEa: 6
- Reviewer 8tHH: 4

---

### Decision · Program_Chairs · 2026-01-26

Reject